# A New Biosynthetic 6-Phytase Added at 500 Phytase Unit/kg Diet Improves Growth Performance, Bone Mineralization, and Nutrient Digestibility and Retention in Weaned Piglets and Growing–Finishing Pigs

**DOI:** 10.3390/vetsci11060250

**Published:** 2024-06-03

**Authors:** Maamer Jlali, Clémentine Hincelin, David Torrallardona, Tania Rougier, Marcio Ceccantini, Sarper Ozbek, Aurélie Preynat, Estelle Devillard

**Affiliations:** 1Adisseo France S.A.S., European Laboratory of Innovation, Science and Expertise, 69190 Saint-Fons, Lyon, France; tania.rougier@adisseo.com (T.R.); sarper.ozbek@adisseo.com (S.O.); estelle.devillard@adisseo.com (E.D.); 2Adisseo France S.A.S., 92160 Antony, France; clementine.hincelin@adisseo.com (C.H.); marcio.ceccantini@adisseo.com (M.C.); aurelie.preynat@adisseo.com (A.P.); 3Institut de Recerca i Tecnologia Agroalimentàries, Animal Nutrition, Ctra. Reus-El Morell km. 3.8, 43120 Constantí, Tarragona, Spain; david.torrallardona@irta.cat

**Keywords:** availability, bone mineralization, growing pigs, performance, phytase, piglets

## Abstract

**Simple Summary:**

Supplementing piglets and growing–finishing pigs’ diets with a new biosynthetic 6-phytase can be an effective replacement for inorganic phosphorus to improve growth performance, feed efficiency, bone mineralization, and availability of nutrients in weaned piglets and growing–finishing pigs. Phytase addition improved the availability of nutrients, especially of phosphorus and calcium and, therefore, growth performance and bone mineralization in weaning and growing–finishing pigs fed diets reduced in digestible phosphorus and calcium. In addition, the present study highlights that the main pathways (fecal and urinary) of phosphorus and calcium excretion in weaned piglets can be influenced not only by dietary levels of these minerals but also by phytase supplementation.

**Abstract:**

Two experiments were performed to evaluate the effect of a biosynthetic 6-phytase added at 500 phytase unit (FTU)/kg diet on growth performance, bone mineralization, and nutrient digestibility and retention in weaned piglets and growing–finishing pigs. Experiments were performed on 90 weaned male and female piglets with an average initial body weight (BW) at 7.7 ± 0.73 kg, 26 days of age) and 300 male and female growing pigs (initial BW: 21.0 ± 3.44 kg) for 43 and 98 days in experiments 1 and 2, respectively. In each experiment, the animals were assigned to one of three treatments according to a randomized complete block design. The treatments consisted of a positive-control (PC) diet formulated to meet nutrient requirements; a negative-control (NC) diet reduced similarly in calcium (Ca) and digestible P by 0.15 and 0.12% points in phases 1 and 2, respectively, in piglets and by 0.14, 0.11, and 0.10% points, respectively, in phases 1, 2, and 3 in growing–finishing pigs, compared with PC diet; and a NC diet supplemented with the new 6-phytase at 500 FTU/kg diet (PHY). The dietary P and Ca depletion reduced (*p* < 0.05) the final BW (−11.9%; −7.8%,), average daily gain (ADG, −17.8%; −10.1%), average daily feed intake (ADFI, −9.9%; −6.0%), gain-to-feed (G:F) ratio (−8.9%; −4.6%), and apparent total tract digestibility (ATTD) of P (−7.7% points; −6.7% points) in nursery piglets and growing pigs, respectively. It also decreased (*p* < 0.001) P and Ca retention by 6.1 and 9.4% points, respectively, in nursery pigs and ash, P, and Ca contents in metacarpal bones by 18.4, 18.4, and 16.8%, respectively, in growing pigs. Compared to animals fed the NC diet, phytase supplementation improved (*p* < 0.001) the final BW (+7.7%; +11.3%), ADG (+12.5%; +15.0%), G:F ratio (+8.4%; +5.8%), ATTD of Ca (+10.8% points; +7.2% points), and ATTD of P (+18.7% points; +16.6% points) in weaned piglets and growing pigs, respectively. In addition, phytase also increased (*p* < 0.001) P and Ca retention by 6.1 and 9.4% points, respectively, in nursery pigs and ash, P, and Ca contents in metacarpal bones by 17.7, 15.0, and 15.2%, respectively, in growing pigs. The final BW, ADG, G:F ratio, and bone traits in animals fed the NC diet supplemented with phytase were comparable to animals fed the PC diet. This finding indicates the ability of this novel biosynthetic phytase to restore performance and bone mineralization by improving the availability of P and Ca in piglets and growing pigs fed P- and Ca-deficient diets.

## 1. Introduction

Phosphorus (P) is an essential nutrient in swine diets that regulates important functions as part of structural compounds in cell membranes and nucleic acids and is involved in bone development and energy metabolism. Next to energy and amino acids, phosphorus is the third most expensive component in diets for monogastric animals. However, up to 80% of total P in plant feedstuffs, mostly used in weaned piglets’ and growing pigs’ diets, is present as phytate-P [1]. It is well recognized that pigs are unable to totally hydrolyze phytate and utilize P from plant feedstuffs due to the low phytase activity in their gastrointestinal tracts. So, to counteract the antinutrient effects of phytate and improve the bioavailability of P from phytate-P and other nutrients and have a positive impact on growth performance, exogenous phytases can be added to swine diets. Phytases have currently become the most used feed enzyme and are introduced into 70% of total swine diets [2]. The first commercial phytase was isolated from *Aspergillus niger* in 1991, and since then, the identification and production of new microbial phytases have greatly increased [3,4]. Phytases can have diverse origins, including plants, animals, and microorganisms [5,6,7]. Phytases can be classified into 3-phytase or 6-phytases according to the position of the phosphate group that will be cleaved from the inositol ring during the initial hydrolysis [8,9]. Phytases can have different properties, including their affinity to phytate, resistance to acidic conditions and protease enzymes in the gastrointestinal tract, the pH required for optimal activity, and thermal stability during feed processing, which in turn, may affect the in vivo efficiency of phytase products [10]. A new biosynthetic bacterial 6-phytase produced by *Trichoderma reesei* was recently developed, and its efficacy has been demonstrated in broilers [11,12]. This novel phytase is characterized by the stability of the enzyme over a wide pH range, high affinity for myo-inositol hexakisphosphate (IP6), and a high intrinsic thermostability [13]. Therefore, the present study aimed to evaluate the effects of this new biosynthetic bacterial 6-phytase on growth performance, bone mineralization, and nutrient digestibility and retention in nursery and growing pigs.

## 2. Materials and Methods

Two studies were performed at the Experimental Farm of the IRTA Animal Nutrition (Mas Bové; Ctra. Reus-El Morell km. 3.8, E-43120 Constantí, Spain). All experimental procedures were approved by IRTA’s Ethical Committee on Animal Experimentation (references: CEEA 11952 and CEEA 11965, respectively).

### 2.1. Animals, Housing, Experimental Design, and Sampling

#### 2.1.1. Experiment 1—Weaned Piglets

A total of 45 entire weaned male and 45 weaned female piglets (initial body weight (BW): 7.7 ± 0.73 kg, 26 days of age) were allocated by initial body weight and sex into 30 pens (15 of males and 15 of females). Each block contained 6 pens (3 pens of males and 3 of females) with 3 piglets per pen. The 6 pens in each block were randomly assigned to one of three dietary treatments using a randomized complete block design based on the initial BWs of the animals, sex, and pen location. Dietary treatments consisted of a positive control (PC) formulated to meet or exceed all essential nutrient requirements according to FEDNA [14], including calcium (Ca = 0.82% and 0.70% for the pre-starter and starter phases, respectively) and digestible P (0.40 and 0.33% for phases 1 and 2, respectively); a negative control (NC) reduced equivalently in Ca and digestible P by 0.15 and 0.12% points for phases 1 and 2, respectively, in comparison with the PC diet; and a NC diet supplemented with this novel phytase at 500 phytase units (FTU)/kg diet (PHY, Rovabio PhyPlus, Adisseo France SAS, Antony, France). Pre-starter and starter experimental diets based on corn and soybean meal were fed from day 0 to 14 and day 15 to 43, respectively (Table 1). Titanium dioxide was incorporated into all the diets of phase 2 at 0.5% as an indigestible marker for the measurement of apparent total tract digestibility (ATTD) of nutrients. Diets were offered as pellets (3 mm) throughout the experimental period. Experimental diets and water were offered for 43 days. Individual body weights (BWs) and feed intake for each pen were recorded on days 0, 14, and 43, and then, the pen’s average initial and final BWs, average daily gain (ADG), average daily feed intake (ADFI), and average gain-to-feed (G:F) ratio between 0–14, 15–43, and 0–43 days of the experiment were calculated. 

On day 22 of the trial, 30 piglets (1 with the heaviest initial body weight from each pen) were transferred to metabolism cages (1 m × 1 m; piglets kept individually and unrestrained) and continued on the same dietary treatments, for 3 days of adaptation (day 22 to day 24), followed by 4 days of separate collection of urine and faeces (day 25 to day 29), after which they were returned to their original pens. Subsequently, piglets with the intermediate and lightest initial body weights from each pen were also moved to the metabolism cages for 3 days of adaptation (day 29 to day 31 and day 36 to day 38, respectively) and 4 days of urine and faeces collection (day 32 to day 36 and day 39 to day 43, respectively). Urine and faeces were collected twice a day, weighed, homogenized, sampled, and kept frozen until further analysis. 

#### 2.1.2. Experiment 2—Growing–Finishing Pigs

A total of 300 pigs ([Large White × Landrace] × Pietrain; 156 entire males and 144 females; 21.0 ± 3.44 kg; 10 weeks of age) were distributed by initial weight and sex into 75 pens (39 pens of males and 36 pens of females). They were randomly assigned to one of three dietary treatments using a randomized complete block design. The dietary treatments consisted of a positive control (PC) formulated to meet or exceed all essential nutrient requirements according to FEDNA [14], including 0.69% Ca and 0.31% digestible P for days 0 to 28, 0.61% Ca and 0.26% digestible P for days 28 to 56, 0.56% Ca, and 0.23% digestible P for days 56 to 98; a negative-control (NC) diet reduced similarly in digestible P and Ca by 0.14, 0.11, and 0.10% points for days 0 to 28, 29 to 56, and 57 to 98, respectively, in comparison with the PC diet; and a NC diet supplemented with this new 6-phytase at 500 FTU/kg diet (PHY, Rovabio PhyPlus, Adisseo France SAS, France, Table 2). Titanium dioxide (0.5%) was added to the grower 1 diets as a marker for digestibility measurements. The individual weights of animals and feed intake for each pen were measured at the beginning of the experiment (day 0; 10 weeks of age), and at days 28, 56, and 98 of the trial. From days 25 to 28 of the trial, fresh faecal samples (250 to 300 g per day per pen) were collected from each pen for 4 consecutive days, then pooled (around 100 g/pen) by pen and stored at −20 °C until further analysis. Additionally, on day 28 of the trial, one pig per pen was euthanized, and its front left hoof was sampled to assess the mineralization of the metacarpal (III) bones. 

### 2.2. Chemical Analysis

Samples of all the experimental diets were analyzed for dry matter, crude fat (CF), ash, gross energy (GE), neutral detergent fibre (NDF), acid detergent fibre (ADF), lignin, nitrogen (N), Ca, total P, phytate-P, and phytase activity. Additionally, freeze-dried faecal samples were analyzed for dry matter, N, Ca, and total P and urine samples were analysed for N, Ca, and total P. Dry matter, N, ash, and CF (ether extract) were analysed according to AOAC methods [15]. In addition, NDF, ADF, and lignin were measured with an Ankom 200/220 Fiber Analyzer (IT-0602-L-10133 Sequential analysis of NDF, ADF, and lignin, Macedon, NY, USA). The GE contents in the experimental diets were determined with an adiabatic bomb calorimeter (IKA, C-400, Janke & Kunkel KG., Staufen I. Br., Germany; DIN 51900). Total P and Ca were analysed by inductively coupled plasma emission spectroscopy (ICP-OES Optima 2100DV, Perkin Elmer Life and Analytical Sciences, Shelton, CT, USA) according to method 984.27, AOAC, [16]). Phytate-P was determined according to Haug and Lantzsch [17]. The activity of phytase in the experimental diets was analysed according to the ISO standard methodology (ISO Standard 300242, [18]). One unit of phytase is defined as the amount of enzyme that liberates one micromole of inorganic orthophosphate from phytic acid per minute at pH 5.5 and 37 °C. For the bone samples, frozen hooves were thawed overnight and autoclaved at 105 °C for 15 min, after which the metacarpal bones were dissected and cleaned of all tissue, including cartilage caps. Clean metacarpal bones were weighed to record fresh weight, and then, they were dried overnight at 103 ± 2 °C, cooled in a dissector, and weighed again to calculate the dry matter. After this, they were incinerated in a muffle furnace at 550 °C for 72 h to obtain ashes.

### 2.3. Calculations and Statistical Analysis

The ATTD of dry matter, crude protein, Ca, and P were calculated according to the following equation:ATTD (%) = (1 − [(Ti_feed_/Ti_faeces_) × (N_faeces_/N_feed_)]) × 100
where Ti_feed_ = TiO_2_ concentration in feed; Ti_faeces_ = TiO_2_ concentration in faeces; N_faeces_ = nutrient (dry matter, crude protein, Ca, or P) concentration in the faeces; and N_feed_ = nutrient concentration in the feed. 

The balance retention of Ca and P were calculated as follows:Balance (%, Ca or P) = [(nutrient intake − (nutrient in faeces + nutrient in urine)/nutrient intake)] × 100

In experiment 1, the pen was considered as the experimental unit for all performance measurements, and the statistical model used to analyze the data was:Y_ij_ = μ + T_i_ + B_j_ + S_k_ + T_i_ × S_k_ + ε_ijk_
where Y_ij_ = observation on ith dietary treatment (T) in jth block (B) and kth sex (S), μ = overall mean, T_i_ = treatment effect, B_j_ = block effect, S_k_ = sex effect, T_i_ × S_k_ = treatment × sex interaction, and ε_ijk_ = experimental error. 

For the digestibility and balance data, individual animals were considered as the experimental unit. The statistical model used to analyse the digestibility and balance data was:Y_ijk_ = μ + T_i_ + B_j_ + S_k_ + P_l_ + T_i_ × S_k_ + T_i_ × P_l_ + S_k_ × P_l_ + ε_ijkl_
where Y_ijk_ = observation on ith dietary treatment (T) in jth block (B), kth sex (S) and lth period/weight category (P) considered as batch effect, μ = overall mean, T_i_ = treatment effect, B_j_ = block effect, S_k_ = Sex effect, P_l_ = period/weight category effect, T_i_ × S_k_ = treatment × sex interaction, T_i_ × P_l_ = treatment × period interaction, S_k_ × P_l_ = sex × period interaction, and ε_ijkl_ = experimental error. 

In experiment 2, the pen was used as the experimental unit, and the statistical model used to analyse all the data was:Y_ijk_ = μ + T_i_ + B_j_ + S_k_ + T_i_ × S_k_ + ε_ijk_
where Y_ijk_ = observation on ith dietary treatment (T) in jth block (B) and Sth sex (S), μ = overall mean, T_i_ = treatment effect, B_j_ = block effect, S_k_ = sex effect, and ε_ijk_ = experimental error.

Data were analysed using the GLM procedure of SAS (SAS Inst. Inc., Cary, NC, USA). Comparisons of least square means for each significant effect were performed by the Tukey–Kramer test. Orthogonal contrasts were done to compare the NC diet vs. the PC diet and the PHY vs. the NC diet. Statistical significance was set at *p* < 0.05, and 0.05 < *p* < 0.10 was considered a trend.

## 3. Results

### 3.1. Experimental Diets

The analysed dietary Ca and total P contents (Table 3 and Table 4) were, in general, in agreement with the expected values. In Exp. 1, the analysed phytase activities in the experimental diets supplemented with phytase were 489 and 440 FTU/kg diet in the pre-starter and starter diets, respectively. In Exp. 2, the analysed phytase activities in the experimental diets supplemented with phytase were 438, 411, and 541 FTU/kg diet in the grower 1, grower 2, and finisher diets, respectively.

### 3.2. Experiment 1—Nursery Piglets

There were no interactions among the main effects (batch, sex, and diet) for any of the variables measured, and thus, only the effects of dietary treatments and sex are presented. Piglets fed the NC diet had reduced (*p* < 0.05) BW, ADG, and G:F ratio compared to the pigs fed the positive-control diet (Table 5). For the global experimental period, the piglets that were fed with the NC diet reduced in digestible P and Ca had lesser final body weight (−11.9%, *p* = 0.001), ADG (−17.8%, *p* = 0.001), ADFI (−9.9%, *p* = 0.04), and G:F ratio (−8.9%; *p* < 0.05) than the piglets that were fed with the PC diet. Phytase supplementation tended to improve the final BW (+7.7%, *p* = 0.07) and ADG (+12.5%, *p* = 0.06) compared to the piglets fed the NC diet. In addition, the piglets offered the NC diet supplemented with phytase had a higher (+8.4%, *p* = 0.008) G:F ratio than piglets fed the NC diet. 

There were no significant differences in growth-performance parameters between piglets fed the NC diet supplemented with phytase and piglets fed the PC diet. As shown in Table 6, digestible P and Ca reduction in NC did not influence the ATTD of dry matter, crude protein, and Ca, while it decreased (*p* < 0.001) the ATTD of P. However, piglet batch influenced (*p* < 0.05) the ATTD of dry matter, crude protein, and Ca and the retention of Ca. Also, the ATTD of Ca was influenced (*p* = 0.02) by sex. However, neither the retention of P nor of P was influenced by sex.

Phytase supplementation increased (*p* < 0.001) the ATTD of Ca and P by 10.8 and 18.7% points, respectively, in comparison to NC. The P intake and P output in faeces and urine were greater (*p* < 0.001) in weaned piglets fed the PC diet than in pigs fed the NC diet (Table 7). 

Compared with piglets on the NC diet, Ca intake and Ca output in faeces were higher (*p* < 0.001), while Ca output in urine was lower (*p* < 0.001) in piglets that were fed the PC diet. Piglets fed the PC diet had greater (*p* < 0.001) retention of P and Ca by 6.1 and 9.4% points, respectively, in comparison with piglets fed the NC diet. Phytase supplementation reduced (*p* < 0.001) the P excretion in faeces and Ca excretion in faeces and urine in comparison to piglets fed the NC diets. Compared with piglets fed the PC diet, piglets fed with the NC diet had a lower (*p* < 0.001) ATTD of P by 7.7% points. However, the supplementation of phytase to the NC diet improved (*p* < 0.001) the ATTD of P and Ca by 18.7 and 10.8% points, respectively.

### 3.3. Experiment 2—Growing–Finishing Pigs

No significant interaction was observed between dietary treatment and sex on the growth-performance parameters, bone traits, and the ATTD of nutrients, and thus, only the effects for sex and dietary treatments are presented. Pigs fed the NC diet reduced in digestible P and Ca had a lower final body weight (−7.8%, *p* < 0.001), ADG (−10.1%, *p* < 0.001), ADFI (−6.0%, *p* = 0.005), and G:F ratio (−4.6%, *p* < 0.001) than the pigs that were fed with the PC diet (Table 8). 

Additionally, the digestible P and Ca reduction in the NC diet decreased (*p* < 0.001) the dry weight, ash content, P content, and Ca content in metacarpal bones in pigs receiving that diet compared to the pigs that were fed the PC diet (Table 9). 

Compared with growing pigs fed with the NC diet, the pigs fed with the NC diet supplemented with phytase at 500 FTU/kg diet had a greater (*p* < 0.001) final body weight (+11.3%), ADG (+15.0%), ADFI (+8.9%), and G:F ratio (+5.8%). In addition, phytase addition to the NC diet improved (*p* < 0.001) dry weight, ash content, P content, and Ca content in the metacarpal bones by 7, 17.7, 15.0, and 15.2%, respectively, in comparison to the NC diet. There were no significant differences between the pigs that were fed the NC diet supplemented with phytase and those that were fed the PC regarding final BW, ADG, ADFI, G: F ratio, bone dry weight, ash content, P content, and Ca content. Regarding the digestibility of nutrients (Table 10), digestible P and Ca reduction increased (*p* < 0.05) the ATTD of Ca by 2.7% points, while it decreased (*p* < 0.001) that of P by 6.7% points in comparison to the PC group. 

The addition of phytase improved (*p* < 0.001) the ATTD of Ca (+7.2% points) and P (+16.6% points) in comparison to the NC diet. No effects of dietary treatment on the ATTD of dry matter and crude protein were observed.

## 4. Discussion

In the experimental diets supplemented with phytase, the phytase activities agreed with the targeted phytase activity (500 FTU/kg diet). They can be considered acceptable, considering the sensitivity and the relative standard deviation of the analysis and the errors introduced by enzyme application, mixing, and sampling. The phytase activities observed in PC and NC diets were in accordance with the plant-feed ingredients (corn, soybean meal, sunflower meal, extruded soybeans, and soy-protein concentrate) usually characterized by their low intrinsic phytase activities. 

The effects of a novel biosynthetic phytase supplemented at 500 FTU/kg diet on growth performance, bone mineralization, and nutrient availability in nursery and growing–finishing pigs were investigated. Exogenous phytases added routinely to swine diets break down phytic acid, thereby increasing the availability of P, improving growth performance, and reducing the excretion of P into the environment [19]. However, the benefits of adding exogenous phytases to livestock feeds, especially in nursery and growing pigs, can mainly depend on their source and dose as well as on the nutritional characteristics of diets. As expected, the dietary reduction in P and Ca reduced growth performance in piglets and growing–finishing pigs in comparison to the animals fed the P- and Ca-adequate diet. Several studies have reported that the growth performance of piglets and growing pigs was reduced when dietary Ca and P were below the requirements [20,21,22,23,24,25,26]. These results indicate the key roles of P and Ca in ensuring the productivity of piglets and growing–finishing pigs. In fact, P is an indispensable nutrient needed for normal growth, and it plays important roles in vital functions, such as energy and amino acid metabolism, protein synthesis, and maintaining acid–base balance [27,28,29,30]. Several studies have reported improvements in performance when phytase was used in piglets and growing pigs [21,31,32,33,34,35]. In the current study, the addition of phytase at 500 FTU/kg diet improved the growth performance and feed efficiency of nursery and growing–finishing pigs fed diets reduced in digestible P and Ca. These findings indicate the capacity of phytase to release P and Ca and probably the other nutrients chelated by the phytate molecule, which therefore, improves the growth performance of nursery and growing pigs. It was confirmed by the results observed on the ATTD of Ca and P. 

Indeed, in the current study, the result that 6-phytase supplementation increased the ATTD of P and Ca in experiments 1 and 2 carried out on weaned piglets and growing pigs, respectively, agrees with the results from previous experiments [26,36,37,38,39]. These improvements in P and Ca digestibility could be explained by the mechanism of action of phytase to release minerals, especially P and Ca, from the phytate bond, as well as limiting the interactions between nutrients and phytate within the gastrointestinal tract [40]. However, Bournazel et al. [41] reported that the addition of a microbial phytase at 500 FTU/kg diet did not influence the ATTD of Ca but increased the retainable Ca in growing pigs. These different outcomes may be due to phytase type and dose, Ca level, and source, as well as the Ca:P ratio, as examples and other factors may influence the response of phytase on the ATTD of Ca. In experiment 1 performed on weaned piglets, the Ca and P balance data showed that most of the P excreted was faecal in the piglets fed an adequate-nutrient diet (96.0%) or a P- and Ca-deficient diet (98.7%). Phytase addition to the P- and Ca-deficient diet reduced faecal loss of P, thereby improving the total P retention, but it had no effect on urinary P loss. Excretion of Ca mostly occurred in faeces (95.3% of total Ca excreted) in piglets fed an adequate-nutrient diet. However, it was split between faecal (79.1% of total Ca excreted) and urinary (20.9% of total Ca excreted) in piglets fed a P- and Ca-deficient diet. Phytase addition reduced the faecal and urinary losses of Ca, and therefore, a greater total calcium retention was observed in piglets fed a P- and Ca-deficient diet supplemented with phytase. These findings indicate that the excretion pathways of P and Ca can be influenced not only by the dietary level of P and Ca but also by phytase supplementation. Phytase seems able to regulate the metabolism of P and Ca by controlling their main excretion pathways. However, the molecular mechanisms by which phytase regulates P and Ca excretion, especially in kidneys, remains unknown and further studies are needed to elucidate them. It is important to indicate that the initial body weights of piglets can influence the ATTD of dry matter, crude protein, and Ca, and Ca retention. There was also a difference between male and female piglets on the ATTD of Ca and Ca outputs in faeces and urine. 

The results on dry weight, ash, P, and Ca contents in the metacarpal bones of growing pigs are in line with the growth-performance and digestibility results. Many studies have demonstrated that dietary Ca and P play an indispensable role in skeletal development and mineralization in pigs [27,28,30]. As observed on growth performance, there was a negative effect of P and Ca reduction on dry weight, ash, P, and Ca contents in the metacarpal bones of growing pigs. This could be an indicator that reducing digestible P and Ca similarly by 0.14, 0.11, and 0.10% points, respectively, in phases 1, 2, and 3 relative to the FEDNA [14] recommended level in growing–finishing pigs was sufficient to induce negative effects not only on growth performance but also on bone mineralization. In agreement with the data generated with other exogenous phytases [42,43], phytase supplementation at 500 FTU/kg diet in P- and Ca-deficient diets improved the ash, P, and Ca contents in the metacarpal bones of growing pigs. The levels of the metacarpal bones’ dry weight, ash, P, and Ca contents in animals fed with the phytase-supplemented diets became equivalent to those fed with the PC-nutrient adequate diets. This observation is the result of the increased ATTD of Ca and P due to phytase supplementation, and it indicates the effectiveness of this new biosynthetic 6-phytase to improve the P and Ca availability from P- and Ca-deficient diets, which are the most abundant minerals in the skeletal [27,44]. However, further studies are needed to elucidate the relationship between phytase, its substrate (phytate), and digestible Ca level on nutrient availability, in addition to P and Ca, such as amino acids and trace minerals, energy utilization, and growth performance in nursery and growing pigs.

## 5. Conclusions

In conclusion, the advantages of phytase addition into swine diets are well-established. The findings of the present study confirmed that reducing the digestible P and Ca similarly by 0.15 and 0.12% points, in phases 1 and 2, respectively, in weaned piglets and 0.14, 0.11, and 0.10% points, in phases 1, 2 and 3, respectively, in growing–finishing pigs relative to the FEDNA recommended levels was able to produce negative effects on growth performance, digestibility, and retention of P and Ca and bone mineralization in nursery and growing pigs. This study also demonstrated that the addition of the new biosynthetic 6-phytase has the potential to improve the digestibility of nutrients and, therefore, the performance and bone mineralization. It also indicates that the main pathways (faecal and urinary) of P and Ca excretion can be influenced not only by the dietary level of these minerals but also by phytase supplementation in piglet diets. 

## Figures and Tables

**Table 1 vetsci-11-00250-t001:** Composition and characteristics of basal diets for weanling piglets (Experiment 1).

	Phase 1 (Pre-Starter, 0 to 14 Day)		Phase 2 (Starter, 15 to 43 Day)	
	PC	NC	PC	NC
Composition, %				
Corn	52.67	52.67	65.16	65.77
Sweet milk whey	10.97	10.97	-	-
Soybean meal, 48% crude protein	17.00	17.00	20.00	20.00
Extruded soybeans	5.98	5.98	6.94	6.79
Soy-protein concentrate	5.00	5.00	-	-
Sugar-beet pulp	3.00	3.80	2.03	2.19
Animal fat	2.00	2.00	1.80	1.80
L-lysine	0.50	0.50	0.56	0.56
L-threonine	0.15	0.15	0.17	0.17
L-tryptophan	0.03	0.03	0.04	0.04
DL-methionine	0.20	0.20	0.17	0.17
L-valine	0.04	0.04	0.05	0.05
Sodium chloride	0.28	0.27	0.46	0.45
Calcium carbonate	0.01	0.34	-	0.28
Dicalcium phosphate	1.74	0.63	1.70	0.81
Titanium dioxide	-	-	0.50	0.50
Premix ^1^	0.40	0.40	0.40	0.40
Noxyfeed ^2^	0.02	0.02	0.02	0.02
Calculated nutrients, %				
NE, kcal/kg	2544	2555	2497	2512
Crude protein	19.64	19.72	17.70	17.71
Crude fat	5.29	5.31	5.68	5.68
Crude fiber	2.72	2.86	2.76	2.79
Ash	5.64	4.89	5.33	4.73
Dig. Lysine	1.35	1.35	1.23	1.23
Dig. Methionine	0.47	0.47	0.42	0.42
Dig. Met + Cys	0.74	0.74	0.68	0.68
Dig. Threonine	0.80	0.80	0.73	0.73
Dig. Tryptophan	0.22	0.22	0.20	0.20
Dig. Valine	0.86	0.86	0.78	0.78
Calcium	0.82	0.67	0.70	0.58
Total phosphorus	0.70	0.51	0.63	0.41
Phytic phosphorus	0.22	0.22	0.24	0.24
Dig. phosphorus	0.40	0.25	0.33	0.21

^1^ Provides per kg feed: vitamin A, 10,000 IU; vitamin D_3_, 2000 UI; vitamin E (α-tocopherol), 25 mg; vitamin B_1_, 1.5 mg; vitamin B_2_, 3.5 mg; vitamin B_6_, 2.4 mg; vitamin B_12_, 20 µg; vitamin K_3_, 1.5 mg; calcium panthotenate, 14 mg; nicotinic acid, 20 mg; folic acid, 0.5 mg; biotin, 50 µg; Fe (from FeSO_4_·H_2_O), 120 mg; I (from Ca(I_2_O_3_)_2_), 0.75 mg; Cu (from CuSO_4_·5H_2_O), 6 mg; Mn (from MnO), 60 mg; Zn (from ZnO), 110 mg; Se (from Na_2_SeO_3_), 0.37 mg. ^2^ Contains BHT+ propyl galate (56%) and citric acid (14%).

**Table 2 vetsci-11-00250-t002:** Composition and characteristics of basal diets for growing–finishing pigs (Experiment 2).

	Grower 1		Grower 2		Finisher Phase	
	PC	NC	PC	NC	PC	NC
Composition, %						
Corn	67.93	69.07	73.39	74.28	79.73	80.54
Soybean meal, 48% crude protein	19.06	19.06	14.61	14.61	6.98	6.98
Sunflower meal	7.23	7.23	8.08	8.08	10.24	10.24
Animal fat	2.35	1.93	1.47	1.14	0.80	0.49
Calcium carbonate	0.10	0.44	0.15	0.41	0.17	0.40
Dicalcium phosphate	1.58	0.54	1.24	0.42	1.07	0.32
L-lysine	0.35	0.34	0.23	0.23	0.22	0.22
L-threonine	0.05	0.04	-	-	-	-
L-tryptophan	0.01	0.01	-	-	-	-
DL-methionine	0.02	0.01	-	-	-	-
Sodium chloride	0.41	0.41	0.41	0.41	0.39	0.39
Premix ^1^	0.40	0.40	0.40	0.40	0.40	0.40
Noxyfeed ^2^	0.02	0.02	0.02	0.02	0.02	0.02
Titanium dioxide	0.50	0.50	-	-	-	-
Calculated nutrients, %						
Net energy, kcal/kg	2475	2475	2475	2475	2475	2475
Crude protein	17.00	17.08	15.55	15.62	13.33	13.39
Crude fat	5.11	4.73	4.36	4.06	3.81	3.54
Crude fiber	3.40	3.42	3.45	3.47	3.60	3.62
Ash	5.23	4.54	4.28	3.74	3.84	3.35
Dig. Lysine	0.98	0.98	0.79	0.79	0.61	0.61
Dig. Methionine	0.28	0.28	0.26	0.26	0.23	0.24
Dig. Met + Cys	0.55	0.55	0.51	0.51	0.46	0.46
Dig. Threonine	0.59	0.59	0.49	0.49	0.40	0.40
Dig. Tryptophan	0.17	0.17	0.14	0.14	0.11	0.11
Calcium	0.69	0.55	0.61	0.50	0.56	0.46
Total phosphorus	0.64	0.46	0.57	0.43	0.53	0.40
Phytic phosphorus	0.27	0.27	0.27	0.27	0.27	0.27
Digestible phosphorus, %	0.31	0.17	0.26	0.15	0.23	0.13

^1^ Provides per kg feed: vitamin A, 5500 IU; vitamin D_3_, 1100 UI; vitamin E (α-tocopherol), 25 mg; vitamin B_1_, 0.5 mg; vitamin B_2_, 1.4 mg; vitamin B_6_, 1 mg; vitamin B_12_, 8 µg; vitamin K_3_, 0.5 mg; calcium panthotenate, 5.6 mg; nicotinic acid, 8 mg; choline, 120 mg; Fe (from FeSO_4_·H_2_O), 80 mg; I (from Ca(I_2_O_3_)_2_), 0.5 mg; Cu (from CuSO_4_·5H_2_O), 10 mg; Mn (from MnO), 40 mg; Zn (from ZnO), 100 mg; Se (from Na_2_SeO_3_), 0.25 mg. ^2^ Contains BHT+ propyl galate (56%) and citric acid (14%). PC: positive-control diet; NC: negative-control diet similarly reduced by 0.14, 0.11, and 0.10% points in digestible phosphorus and calcium in grower 1, grower 2 and finisher diets, respectively.

**Table 3 vetsci-11-00250-t003:** Analysed nutrients of experimental diets (Experiment 1).

Item	Phase 1 (Pre-Starter, 0 to 14 d)			Phase 2 (Starter, 15 to 43 d)		
	PC	NC	PHY	PC	NC	PHY
Dry matter, %	89.30	88.90	89.20	88.60	89.00	88.90
Crude protein, %	19.30	19.30	19.40	17.60	17.20	17.60
Crude fat, %	5.50	5.50	5.50	5.70	5.80	5.80
Ash, %	5.20	4.60	4.60	4.90	4.50	4.50
GE, kcal/kg	4076	4086	4094	4046	4095	4096
Ca, %	0.72	0.58	0.58	0.63	0.51	0.52
Total P, %	0.64	0.47	0.46	0.6	0.46	0.46
Phytase activity, FTU/kg	59	56	489	139	96	440

PC: positive-control diet; NC: negative-control diet equivalently reduced by 0.15 and 0.12% points of digestible P and Ca in phases 1 and 2, respectively; PHY: negative-control diet supplemented with phytase at 500 FTU/kg diet. GE: gross energy; Ca: calcium; P: phosphorus.

**Table 4 vetsci-11-00250-t004:** Analysed nutrients of experimental diets (Experiment 2).

	Grower 1			Grower 2			Finisher		
	PC	NC	PHY	PC	NC	PHY	PC	NC	PHY
Dry matter, %	87.90	88.00	87.50	88.00	87.20	87.80	87.60	87.20	87.30
Crude protein, %	16.70	16.90	16.60	15.10	15.00	15.10	12.60	12.70	12.70
Crude fat, %	5.10	4.90	4.70	4.00	3.60	3.60	3.60	3.30	3.20
Ash, %	4.90	4.40	4.40	4.00	3.50	3.60	3.40	3.10	3.20
GE, kcal/kg	3946	3949	3943	3935	3886	3906	3859	3871	3879
Total P, %	0.65	0.46	0.46	0.54	0.41	0.41	0.50	0.39	0.39
Ca, %	0.70	0.51	0.53	0.56	0.44	0.46	0.52	0.43	0.41
Phytate-P, %	0.24	0.22	0.23	0.23	0.25	0.23	0.21	0.20	0.20
Phytase, FTU/kg	100	133	438	66	64	411	71	81	541

PC: positive-control diet; NC: negative-control diet equivalently reduced by 0.14, 0.11 and 0.10% points in digestible P and Ca in grower 1, grower 2 and finisher diets, respectively; PHY: negative-control diet supplemented with phytase at 500 FTU/kg diet; GE: gross energy; Ca: calcium; P: phosphorus.

**Table 5 vetsci-11-00250-t005:** Growth performance of weaning piglets fed corn–soybean-meal-based diets without or with phytase for 43 days (Experiment 1).

Item	Treatment			SEM	*p*-Value			
	PC	NC	PHY		Sex	Treatment	PC vs. NC	PHY vs. NC
Phase 1, 0–14 days								
BW day 0, kg	7.71	7.75	7.72	0.083	0.804	0.522	0.537	0.646
BW day 14, kg	10.61 ^a^	9.99 ^b^	10.27 ^ab^	0.521	0.498	0.044	0.035	0.456
ADG 1–14 days, g/d	207 ^a^	160 ^b^	182 ^ab^	38.10	0.482	0.036	0.028	0.395
ADFI 1–14 days, g/d	254	231	237	38.30	0.576	0.397	0.388	0.931
G:F ratio 1–14 days	0.806 ^a^	0.689 ^b^	0.766 ^a^	0.071	0.502	0.004	0.003	0.057
Phase 2, 15–43 days								
BW day 43, kg	23.63 ^a^	20.81 ^b^	22.41 ^ab^	1.516	0.624	0.002	0.001	0.067
ADG 14–43 days, g/d	449 ^a^	373 ^b^	419 ^ab^	42.20	0.374	0.002	0.002	0.060
ADFI 14–43 days, g/d	701 ^a^	632 ^b^	656 ^ab^	53.90	0.236	0.026	0.022	0.589
G:F ratio 14–43 days	0.640 ^a^	0.591 ^b^	0.638 ^a^	0.040	0.917	0.017	0.030	0.036
Global period								
ADG 1–43 days, g/d	370 ^a^	304 ^b^	342 ^ab^	35.10	0.637	0.001	0.001	0.060
ADFI 1–43 days, g/d	556 ^a^	501 ^b^	519 ^ab^	45.80	0.429	0.042	0.036	0.658
G:F ratio 1–43 days	0.665 ^a^	0.606 ^b^	0.657 ^a^	0.033	0.694	0.001	0.002	0.006

^a,b^ Means without a common superscript letter within a row differ (*p* < 0.05). PC: positive-control diet; NC: negative-control diet equivalently reduced by 0.15 and 0.12% points of digestible P and Ca in phases 1 and 2, respectively; PHY: negative-control diet supplemented with phytase at 500 FTU/kg diet; SEM: standard error of the mean.

**Table 6 vetsci-11-00250-t006:** Apparent total tract digestibility of dry matter, crude protein, calcium, and phosphorus in weaned piglets fed corn–soybean-meal-based diets without and with phytase (Experiment 1).

Item	Treatment			SEM	*p*-Value				
	PC	NC	PHY		Batch ^1^	Sex	Treatment	PC vs. NC	PHY vs. PC
Dry matter, %	87.75	87.41	88.22	1.350	0.021	0.805	0.071	0.588	0.057
Crude protein, %	82.91	81.43	82.95	3.030	0.001	0.495	0.094	0.148	0.138
Ca, %	48.81 ^b^	49.81 ^b^	60.65 ^a^	5.800	0.010	0.024	<0.001	0.782	<0.001
P, %	43.26 ^b^	35.53 ^c^	54.27 ^a^	4.860	0.128	0.829	<0.001	<0.001	<0.001

^a,b,c^ Means without a common superscript letter within a row differ (*p* < 0.05). ^1^ Batch: piglets with the highest, intermediate, and lightest body weights transferred to metabolism cages to determine apparent total tract digestibility and phosphorus and calcium balance. PC: positive-control diet; NC: negative-control diet equivalently reduced by 0.15 and 0.12% points of digestible P and Ca in phases 1 and 2, respectively; PHY: negative-control diet supplemented with phytase at 500 FTU/kg diet; SEM: standard error of the mean.

**Table 7 vetsci-11-00250-t007:** Phosphorus (P) and calcium (Ca) balance in weaned piglets fed corn–soybean-meal-based diets without and with phytase (Experiment 1).

Item	Treatment			SEM	*p*-Value				
	PC	NC	PHY		Batch ^1^	Sex	Treatment	PC vs. NC	PHY vs. NC
P intake, g/d	3.79 ^a^	2.86 ^b^	2.90 ^b^	0.218	<0.001	0.059	<0.001	<0.001	0.801
P in faeces, g/d	2.15 ^a^	1.84 ^b^	1.32 ^c^	0.193	<0.001	0.220	<0.001	<0.001	<0.001
P in urine, g/d	0.089 ^a^	0.025 ^b^	0.040 ^b^	0.044	0.228	0.341	<0.001	<0.001	0.378
P retained, g/d	1.55 ^a^	1.00 ^b^	1.54 ^a^	0.202	<0.001	0.277	<0.001	<0.001	<0.001
P retention, %	40.72 ^b^	34.64 ^c^	52.84 ^a^	5.430	0.064	0.863	<0.001	<0.001	<0.001
Ca intake, g/d	3.99 ^a^	3.19 ^b^	3.23 ^b^	0.234	<0.001	0.058	<0.001	<0.001	0.789
Ca in faeces, g/d	2.04 ^a^	1.59 ^b^	1.26 ^c^	0.233	<0.001	0.007	<0.001	<0.001	<0.001
Ca in urine, g/d	0.10 ^c^	0.42 ^a^	0.23 ^b^	0.120	0.043	0.022	<0.001	<0.001	<0.001
Ca retained, g/d	1.85 ^a^	1.18 ^b^	1.74 ^a^	0.252	<0.001	0.724	<0.001	<0.001	<0.001
Ca retention, %	46.19 ^b^	36.82 ^c^	53.64 ^a^	6.555	0.032	0.555	<0.001	<0.001	<0.001

^a,b,c^ Means without a common superscript letter within a row differ (*p* < 0.05). PC: positive-control diet; NC: negative-control diet equivalently reduced by 0.15 and 0.12% points of digestible P and Ca in phases 1 and 2, respectively; PHY: negative-control diet supplemented with phytase at 500 FTU/kg diet; SEM: standard error of the mean. ^1^ Batch: piglets with highest, intermediate, and lightest body weights transferred to metabolism cages to determine apparent total tract digestibility and phosphorus and calcium balance.

**Table 8 vetsci-11-00250-t008:** Performance of growing–finishing pigs fed corn–soybean-meal-based diets without and with phytase (Experiment 2).

Item	Treatment			SEM	*p*-Value			
	PC	NC	PHY		Sex	Treatment	PC vs. NC	PHY vs. NC
Phase 1, 0–28 days								
Initial weight, kg	21.13	21.13	21.2	0.445	0.028	0.814	0.999	0.855
Final weight, kg	39.02 ^a^	37.34 ^b^	38.75 ^a^	1.709	0.701	0.002	0.003	0.014
ADG, g/d	639 ^a^	579 ^b^	627 ^a^	57.60	0.309	0.001	0.002	0.013
ADFI, g/d	1229	1192	1233	96.20	0.579	0.250	0.361	0.284
G:F ratio	0.520 ^a^	0.485 ^b^	0.509 ^a^	0.020	0.001	<0.001	<0.001	0.001
Phase 2, 29–56 days								
Final weight, kg	58.97 ^a^	55.24 ^b^	59.28 ^a^	2.922	0.721	<0.001	<0.001	<0.001
ADG, g/d	713 ^a^	639 ^b^	733 ^a^	61.80	0.818	<0.001	0.001	<0.001
ADFI, g/d	1731 ^a^	1634 ^b^	1778 ^a^	109.00	0.446	<0.001	0.007	<0.001
G:F ratio	0.412 ^a^	0.391 ^b^	0.414 ^a^	0.019	0.120	<0.001	0.001	<0.001
Phase 3, 57–98 days								
Final weight, kg	90.94 ^a^	83.86 ^b^	93.31 ^a^	5.562	0.057	<0.001	<0.001	<0.001
ADG, g/d	761 ^a^	681 ^b^	810 ^a^	75.90	0.003	<0.001	0.001	<0.001
ADFI, g/d	2236 ^a^	2073 ^b^	2302 ^a^	178.20	0.021	<0.001	0.006	<0.001
G:F ratio	0.340 ^ab^	0.328 ^b^	0.352 ^a^	0.019	0.054	0.001	0.071	<0.001
Overall period, 0–98 days								
ADG, g/d	712 ^a^	640 ^b^	736 ^a^	55.70	0.034	<0.001	<0.001	<0.001
ADFI, g/d	1804 ^a^	1696 ^b^	1847 ^a^	117.80	0.232	<0.001	0.005	<0.001
G:F ratio	0.395 ^a^	0.377 ^b^	0.399 ^a^	0.014	0.017	<0.001	<0.001	<0.001

^a,b^ Means without a common superscript letter within a row differ (*p* < 0.05). PC: positive-control diet; NC: negative-control diet equivalently reduced by 0.14, 0.11 and 0.10% points in digestible P and Ca in grower 1, grower 2, and finisher diets, respectively; PHY: negative-control diet supplemented with phytase at 500 FTU/kg diet; SEM: standard error of the mean.

**Table 9 vetsci-11-00250-t009:** Bone traits in growing pigs fed diets without or with phytase (Experiment 2).

Item	Treatment			SEM	*p*-Value			
	PC	NC	PHY		Sex	Treatment	PC vs. NC	PHY vs. NC
Bone dry weight, g/bone	6.58 ^a^	5.98 ^b^	6.40 ^a^	0.57	0.138	0.002	0.002	0.036
Bone ash, g/bone	2.56 ^a^	2.09 ^b^	2.46 ^a^	0.21	0.913	<0.001	<0.001	<0.001
Bone ash, %	39.0 ^a^	35.1 ^b^	38.5 ^a^	2.24	0.001	<0.001	<0.001	<0.001
Bone P, g/bone	0.49 ^a^	0.40 ^b^	0.46 ^a^	0.04	0.888	<0.001	<0.001	<0.001
Bone P, %	7.40 ^a^	6.61 ^b^	7.20 ^a^	0.45	0.008	<0.001	<0.001	<0.001
Bone Ca, g/bone	0.95 ^a^	0.79 ^b^	0.91 ^a^	0.08	0.650	<0.001	<0.001	<0.001
Bone Ca, %	14.5 ^a^	13.2 ^b^	14.2 ^a^	0.92	0.004	<0.001	<0.001	0.002

^a,b^ Means without a common superscript letter within a row differ (*p* < 0.05). PC: positive-control diet; NC: negative-control diet equivalently reduced by 0.14, 0.11, and 0.10% points in digestible P and Ca in grower 1, grower 2, and finisher diets, respectively; PHY: negative-control diet supplemented with phytase at 500 FTU/kg diet; SEM: standard error of the mean.

**Table 10 vetsci-11-00250-t010:** Apparent total tract digestibility of dry matter, crude protein, calcium, and phosphorus in growing pigs fed corn–soybean-meal-based diets without and with phytase (Experiment 2).

Item	Treatment			SEM	*p*-Value			
	PC	NC	PHY		Sex	Treatment	PC vs. NC	PHY vs. NC
Dry matter, %	85.55	85.20	85.70	1.170	0.005	0.304	0.536	0.291
Crude protein, %	79.98	79.56	79.74	2.190	0.007	0.796	0.779	0.954
Ca, %	52.37 ^c^	55.08 ^b^	62.28 ^a^	3.970	0.757	<0.001	<0.001	<0.001
P, %	45.46 ^b^	38.80 ^c^	55.43 ^a^	4.140	0.359	<0.001	0.049	<0.001

^a,b,c^ Within a row, means without a common superscript letter differ (*p* < 0.05). PC: positive-control diet; NC: negative-control diet similarly reduced by 0.14, 0.11, and 0.10% points in digestible P and Ca in grower 1, grower 2, and finisher diets, respectively; PHY: negative-control diet supplemented with phytase (Rovabio PhyPlus) at 500 FTU/kg diet; SEM: standard error of the mean.

## Data Availability

Data can be made available via email to Maamer Jlali, maamer.jlali@adisseo.com.

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
