# Peer review of "A New Biosynthetic 6-Phytase Added at 500 Phytase Unit/kg Diet Improves Growth Performance, Bone Mineralization, and Nutrient Digestibility and Retention in Weaned Piglets and Growing–Finishing Pigs"

_vetsci, 2024, doi:10.3390/vetsci11060250_

Round 1

Reviewer 1 Report

Comments and Suggestions for Authors

The research paper examines the impact of a novel biosynthetic 6-phytase on the growth performance and availability of calcium (Ca) and phosphorus (P) in piglets and growing-finishing pigs. The study shows that the phytase can be added to pig diets at a level of 500 FTU/kg to enhance performance and support bone mineralization in Ca- and P-deficient diets, ultimately reducing feed costs and nutrients waste.

The manuscript is well written. However, there are some areas require attention, it is necessary to explain the basis for the setting of Ca and P levels in the NC diet in Methods section and reason for the difference of the phytase activities between the calculated and analyzed values in the phytase supplemented diet in Discussion part. Other modifications are needed as outlined below:

Page 4 line 141-142, was the indicator method used? And how much feces was collected?

Page 7 line 238-240, the content described cannot be found in Table 5, in terms of P=0.07...p=0.06... p=0.008...

Page 10 line 298, the title in Table 9 is written incorrect.

Comments on the Quality of English Language

The English writing level of the manuscript is good.

Author Response

Thank you so much for your comments. Please find in attached my answers.

Reviewer 2 Report

Comments and Suggestions for Authors

The manuscript describes two experiments investigating a novel phytase for nursery and finishing swine. The manuscript is not well written and contains confusing sentence structure and inaccurate descriptions of the results. Although the project conclusions appear to have merit, it is somewhat difficult to assess with the manuscript as written.

Lines 59-60: requires rewording to clearly separate the structural and physiological importance of phosphorus

Line 68 and throughout: replace "they" with "phytases"

Line 109: While feed intake may be controlled in individual feeding situations, the manuscript describes pen feeding. Was the feed provided to the pen controlled, but not the feed intake of individual pigs?

Line 135: incomplete; no information after "by"

Lines 140-141: Was feed provided adjusted? 

Lines 185-186, 195-196, 203-204: delete

Lines 206-207: provide details about what contrasts were tested and how tested

Lines 211-221: delete "The analyzed values of dietary nutrients and phytase activities in experimental diets of experiments 1 and 2 are presented in Tables 3 and 4, respectively."

Lines 232-233: delete"There was a significant effect of dietary treatment on growth performance from phase 1
(Table 5)."

Line 234: replace "that were" with "pigs"

Line 236: state "data not reported" for final body weight or include the data

Line 237: check p-value because the table is different

Line 238: state "data not reported" or include the data

Line 239: identify the comparison

Line 240: What was the statistical method to determine the comparisons?

Lines 247-249: inaccurate; reword

Lines 260-261: delete "The P and Ca balance data are shown in Table 7."

Line 284: the "higher F:G ratio (+4.6%, P < 0.001)" is inaccurate

Lines 304-306: move forward; associated with Table 8

Line 328: delete "were being" after "phytases"

Line 329: delete "to" before "breakdown"

Line 353: delete "on phytate with probably efficient" after "phytase"

Line 358: replace "which all these" with "as examples and other"

Lines 325-392: The Discussion section should include more discussion of the results of the study.

Author Response

Thank you so much for your comments. Please find attached my answers.

Reviewer 3 Report

Comments and Suggestions for Authors

The article is very well written, clear and objective.

We only have two considerations to make:

To improve the characterization of the NC + phytase treatment, we suggest in the two experiments, lines 101 to 102 and 136 to 138, to define the NC + phytase treatment as PHY

For explaining how urine was collected from females in metabolic cages

Author Response

(The authors gave the same response as above.)

Reviewer 4 Report

Comments and Suggestions for Authors

GENERAL COMMENTS: The authors investigated the effects of a novel biosynthetic phytase on performance, mineral retention in bones, and nutrient digestibility in pigs. As the price of inorganic phosphate source is quite high and the phosphorus is one of the major environmental concerns in swine production industry, research on phytase is meaningful. The experiment was well designed and the manuscript reads well. Please consider my comments to improve the quality of the manuscript.

1. Throughout the manuscript, feed-to-gain ratio should be replaced with gain:feed. The result descriptions also need to be updated.

2. In Introduction section, the author stated that the novel 6-phytase has been previously examined in broilers. So then, the fact that no study on the novel 6-phytase was reported in pigs can support the originality of this study? The authors may want to emphasize that data for this specific phytase in very limiting in pigs to justify the objectives.

3. In Abstract and Result sections, the authors provide the P-values for comparisons based on Tukey-Kramer, which may be acceptable. However, readers may look for the p-values in the result tables. The authors may simply change the p-values to “P < 0.05” in the result descriptions.

4. While the piglets had ad libitum access to feeds in Exp 1, the feed intake was controlled in Exp. 2. Any specific reason? May want to describe the reasons in the manuscript, perhaps in the M&M or discussion section.

5. In this study, the efficacy of novel phytase source produced by Trichoderma reesei was tested, so I think it is necessary to compare the efficacy of novel source with conventional sources previously tested in literature in Discussion section.

6. In several response criteria, the sex effects are very highly significant. The authors may want to discuss the sex effects in Discussion section.

7. lower case, italic “p” for p-values.

8. In Exp 2, the initial BW may have affected the performance (Table 8). May want to consider the initial BW as a covariate for the statistical analysis.

9. Check the word count limit for the abstract. 200 words?

SPECIFIC COMMENTS:

Title: FTU >> phytase unit

Simple summary

L 20: P >> phosphorus

Abstract: Please define abbreviations (FTU, ADG, ADFI, and ATTD) at their first appearance.

L 32: Please define “BW” to “body weight (BW)”

L 33: Please insert a space to make “BW: 21.0”

L 40, 42: Please insert a space to make “P < 0.001”

L 41: “P = P < 0.001” typo??

L 47: As previously mentioned, “F:G” should be replaced with gain:feed here and throughout the manuscript.

L 56: Please list the keywords in alphabetical order.

Introduction

L 78: 6-Phytase >> 6-phytase

Materials and methods

L 92: Please define “BW” to body weight (BW)

L 97: Please define “Ca” to calcium (Ca)

L 110-114: During the period for digestibility test, the feed allowance was adjusted? If so, what was the basis for the feed allowance?

L 111: Please insert a space to make “1 m”

L 113-118: To be consistent, please replace “d” with “day” or “days” here and in other places. e.g., days 25 to 29

L 131-132 and 135: “PC” and “NC” were already defined in experiment 1.

L 137: phytase unit (FTU) >> FTU

L 143-144: Please provide the information on bone sample preparation for measurements.

L 158: calcium >> Ca

L 160: Do not begin a sentence with an abbreviation.

L 174: DM >> dry matter; CP >> crude protein

L 182: Here and throughout the manuscript, please replace ‘gender’ with ‘sex’

L 187: Please insert a comma to make “… balance data, the individual …”

L 189: In Yijk, use subscript for “ijk”

Results

L 233: Please define “ADG” and “F:G” (expected to be changed to gain:feed)

L 236: Please define “ADG”

L 237: Please delete “(P < 0.05)”

Discussion

L 373: Please change to “ … dry weight, ash, P, and Ca contents in …”

Table 1: In footnote 1, please use subscripts for vitamin number (e.g., 1 for B1).

In footnote 1, please provide the forms of trace minerals (e.g., Cu, 6 mg as copper hydroxychloride?).

L 151: P >> phosphorus; Ca >> calcium

Table 2: Soybean 48% >> Soybean meal, 48% crude protein

NE >> Net energy

In footnote 1, please use subscripts for vitamin number (e.g., 1 for B1).

In footnote 1, please provide the forms of trace minerals (e.g., Cu, 10 mg as copper hydroxychloride?).

Table 3: Dry Matter >> Dry matter; Crude Protein >> Crude protein

Table 5: Please provide SEM instead of RMSE and use the decimal units for SEM identical to those for means consistently.

Please use superscript for a and b.

Please provide 3 decimal units for P-values.

May want to change “d” to “day” or “days” as mentioned in L 113-118.

Table 6: In the title, please delete “(ATTD)”

Would be better to spell out “Ca” and “P” in both title and Table.

Please use the decimal units for SEM identical to those for means consistently.

Please use superscript for a and b.

Please provide 3 decimal units for P-values.

Table 7: Please provide SEM instead of RMSE and use the decimal units for SEM identical to those for means consistently.

Please use superscript for a and b.

Please provide 3 decimal units for P-values.

Table 8: Please use the decimal units for SEM identical to those for means consistently.

Please use superscript for a and b.

Please provide 3 decimal units for P-values.

Please define ADG and ADFI in a footnote.

Table 9: In the title, please delete “nitrogen” and change to “… of phosphorus (P) and calcium (Ca) in …”

Please use the decimal units for SEM identical to those for means consistently.

Please use superscript for a and b.

Please provide 3 decimal units for P-values.

Table 10: In the title, please delete “(ATTD)”

Would be better to spell out “Ca” and “P” in both title and Table.

Please use superscript for a and b.

Please provide 3 decimal units for P-values.

Author Response

(The authors gave the same response as above.)

Round 2

Reviewer 2 Report

Comments and Suggestions for Authors

The manuscript describes the effect of a novel phytase for nursery and grow-finish pigs. Although the manuscript has improved, there remains a number of questions and concerns.

Line 29: replace "They" with "Experiments"

Line 32: insert "one of" before "three"

Line 69: replace "They" with "Phytases"

Line 72: replace "They" with "Phytases"

Line 77: replace "It" with "This novel phytase"

Line 77: replace "with a large pH profile" with "by stability of the enzyme over a wide pH range"

Line 83: What are the details of the housing facility?

Line 89: Were the males intact of castrate?

Line 89: replace "90" with "45" and insert "45 weaned" before "females"

Lines 90-91: replace "10 blocks (5 of males and 5 of females)" with "30 pens (15 of males and 15 of females)"

Line 91: Each block should contain 6 pens - 3 pens male and 3 pens female. Was barn location considered in block? Were there other considerations for blocks?

Line 91: insert "of each sex" after "pens"

Line 100: delete "were" after "diets"

Line 101: replace "and their composition and nutrient characteristics of the experimental diets are presented" with "were fed from day 0 to 14 and day 15 to 43, respectively (Table 1)."

Line 102: delete "in Table 1"

Line 104: What were the characteristics of the feeding schedule and feeders?

Line 106: How was feed intake determined?

Line 118: Was the phytase added on top of NC or formulated into a PHY diet?

Table 1: replace "48%" with ", 48% crude protein"

Line 127: replace "blocks" with "pens"

Line 127: Each block should consist of 6 pens - 3 of male and 3 of female.

Line 128: insert "of each sex" after "pens"

Line 135: delete "phytase units"; use previously defined abbreviation

Line 136: insert "; Table 2" after "France"

Lines 137-138: delete "The composition and calculated nutrient contents of the basal diets are shown in Table 2."

Line 139: replace "controlled" with "measured"

Lines 144-149: move to 'Chemical analysis'

Table 2: the diet formulation for PHY is missing

Line 162: insert "(CF)" after "crude fat"

Line 166: replace "crude fat (ether extract)" with "CF"

Line 167: delete "the" before "AOAC"

Line 167: insert "methods" after "AOAC"

Line 168: provide manufacturer details

Line 169: delete "Gross Energy"; use previously defined abbreviation

Line 171: provide manufacturer details

Line 232: Batch must be defined and explained in Materials and Methods

Table 6: footnote explanation of Batch should be included

Lines 259-262: The significant Ca, % difference by sex result should be presented.

Line 274: insert "pigs fed PC" after "urine in"

Lines 329-401: The discussion is lacking anything about the significant sex and batch effects.

Lines 331-334: Were the levels of phytase in the PC diet expected?

Author Response

Thank you so much for your comments and questions. Please find in attached our answers. 

Best regards.

Round 3

Reviewer 2 Report

Comments and Suggestions for Authors

This manuscript describing the effect of a novel phytase for nursery and grow-finish pigs is much improved compared to previous versions. There are no additional questions or concerns from my review.

Author Response

Dear all,

Thank you so much for your comments and suggestions. Please find attached the final version of manuscript.

Best regards.

Maamer Jlali
